# Oxidative Stress as a Regulatory Checkpoint in the Production of Antiphospholipid Autoantibodies: The Protective Role of NRF2 Pathway

**DOI:** 10.3390/biom13081221

**Published:** 2023-08-05

**Authors:** Maurizio Sorice, Elisabetta Profumo, Antonella Capozzi, Serena Recalchi, Gloria Riitano, Benedetta Di Veroli, Luciano Saso, Brigitta Buttari

**Affiliations:** 1Department of Experimental Medicine, Sapienza University of Rome, 00161 Rome, Italy; maurizio.sorice@uniroma1.it (M.S.); antonella.capozzi@uniroma1.it (A.C.); serena.recalchi@uniroma1.it (S.R.); gloria.riitano@uniroma1.it (G.R.); 2Department of Cardiovascular and Endocrine-metabolic Diseases and Aging, Istituto Superiore di Sanità, 00161 Rome, Italy; elisabetta.profumo@iss.it (E.P.); benedetta.diveroli@guest.iss.it (B.D.V.); 3Department of Physiology and Pharmacology “Vittorio Erspamer”, Sapienza University of Rome, 00185 Rome, Italy; luciano.saso@uniroma1.it

**Keywords:** oxidative stress, antiphospholipid autoantibody, food supplements

## Abstract

**Highlights:**

**Abstract:**

Oxidative stress is a well-known hallmark of Antiphospholipid Antibody Syndrome (APS), a systemic autoimmune disease characterized by arterial and venous thrombosis and/or pregnancy morbidity. Oxidative stress may affect various signaling pathways and biological processes, promoting dysfunctional immune responses and inflammation, inducing apoptosis, deregulating autophagy and impairing mitochondrial function. The chronic oxidative stress and the dysregulation of the immune system leads to the loss of tolerance, which drives autoantibody production and inflammation with the development of endothelial dysfunction. In particular, anti-phospholipid antibodies (aPL), which target phospholipids and/or phospholipid binding proteins, mainly β-glycoprotein I (β-GPI), play a functional role in the cell signal transduction pathway(s), thus contributing to oxidative stress and thrombotic events. An oxidation–antioxidant imbalance may be detected in the blood of patients with APS as a reflection of disease progression. This review focuses on functional evidence highlighting the role of oxidative stress in the initiation and progression of APS. The protective role of food supplements and Nuclear Factor Erythroid 2-Related Factor 2 (NRF2) activators in APS patients will be summarized to point out the potential of these therapeutic approaches to reduce APS-related clinical complications.

## 1. Introduction

Antiphospholipid syndrome (APS) is a systemic autoimmune disease characterized by arterial and venous thrombosis and/or pregnancy morbidity, associated with circulating “anti-phospholipid antibodies” (aPLs), such as lupus anticoagulant (LAC), anticardiolipin antibodies (aCL) and anti-β2-glycoprotein I antibodies (aβ2GPI). Such manifestations can be present in primary APS or associated with an autoimmune systemic disease, such as systemic lupus erythematosus, SLE (secondary APS) [1,2,3]. Other manifestations, including thrombocytopenia, cardiac dysfunction [4], accelerated atherosclerosis, nephropathy, movement disorders and cognitive decline may appear in APS patients [5]. 

A ‘two hit hypothesis’ has been suggested to explain the pathogenesis of APS. The presence of circulating aPLs that destroy the integrity of the endothelium inducing a procoagulant phenotype represents the “first hit”, but aPLs alone are not enough to cause thrombosis, which takes place only in the presence of the “second hit”, a triggering factor which is usually represented by smoking, acute infections, oxidative stress (OS) or inflammation [6,7].

Oxidative stress is considered a key element driving pathophysiological processes that play a role in the onset and progression of various non-communicable diseases. According to its widely endorsed definition, OS arises from an imbalance between oxidants and antioxidants in favor of the oxidants.

When ROS (Reactive Oxygen Species) production increases or their scavenging by antioxidants decreases, cells undergo a process of oxidative stress. ROS are oxygen-containing molecules formed by reduction/oxidation reactions (redox reactions) or electronic excitation. Key ROS molecules include hydroxyl and superoxide free radicals and nonradical molecules, such as hydrogen peroxide. Several cytokines and growth factors regulate the ROS production in the mitochondria, mainly via the electron transport chain, where oxygen is reduced to form superoxide anion [8] peroxisomes (through the β-oxidation of fatty acids) [9] and endoplasmic reticulum (through the oxidation of proteins) [10]. Exposure to exogenous agents, including radiation, heavy metals, atmospheric pollutants and various xenobiotics and chemotherapeutics, leads to the increased production of ROS [11]. Although cytotoxic, ROS are crucial for cellular life and their production in the mitochondria is regulated by several growth factors and cytokines. At a moderate concentration, ROS act as second messengers in the transduction of extracellular signals and in the control of gene expression related to cellular proliferation, differentiation and survival [12]. At higher levels, ROS are also produced by cells as defensive molecules against pathogens [13,14,15]. Excessively high cellular levels of ROS can cause damage to proteins, nucleic acids, lipids, membranes and organelles, which may lead to the activation of such cell death processes as apoptosis [16]. Several lines of evidence prove that ROS can cause DNA damage and contribute to the occurrence of oncogenic mutations [17]. In response to stress, a variety of molecular pathways become activated, including those resulting in an overproduction of reactive ROS, inflammatory signaling and apoptotic cell death. Among the “survival” signaling factors [18], the transcription factor Nuclear Factor Erythroid 2-Related Factor 2 (NRF2) contributes to anti-inflammatory and antioxidant processes and thereby prevents cell death by regulating the expression of phase II detoxifying enzymes, including NAD(P)H quinine oxido-reductase 1 (NQO1), glutathione peroxidase, glutathione glutamate-cysteine ligase (GCL), thioredoxin reductase 1 and heme oxygenase-1, etc., [19,20,21,22].

Oxidative stress largely contributes to APS pathogenesis and its complications. In APS patients, OS favors endothelial dysfunction, mainly associated with the alteration of the NO metabolism, stimulating a prothrombotic and proinflammatory status. Several mechanisms have been reported to explain the role of aPLs as a key promoter of oxidative stress and mitochondrial dysfunction. Several studies describe oxidative stress as a possible source of antigenic epitopes responsible for aPL’s subpopulation appearance. The pathogenic role of oxidative stress in APS is also related to its ability to determine protein structural modifications [23] that, causing the appearance of neo or cryptic epitopes, may sustain the activation of autoimmune reactions and the occurrence of complications in this disease. The different expression of autoantigens might be responsible for the different clinical manifestations of the syndrome.

In this review, we examine the available evidence for the involvement of aPLs in the induction of a pro-oxidative state in APS patients and the possible use of oxidative stress biomarkers in the patient management. Furthermore, we will discuss the in vivo and in vitro evidence on the role of oxidative stress in the post-translational modifications of antigens associated with APS that suggest oxidative stress as a regulatory checkpoint in antiphospholipid autoantibody production. Moreover, the importance of oxidative stress in the activation of multiple signaling pathways that accelerate the progression and exacerbation of the symptoms of APS will be examined. The protective role of food supplements and NRF2 activators in APS patients will be summarized.

## 2. aPL as a Trigger for a Pro-Oxidative State in APS Patients

The role of oxidative stress in the pathogenesis of APS has been widely demonstrated by experimental and clinical studies. It is considered a “second hit” contributing to the induction of the procoagulant phenotype and to the occurrence of thrombosis [6,7,24]. Reactive oxygen species are responsible for lipid peroxidation and the formation of oxidized low-density lipoproteins (oxLDL). oxLDL interact with β2GPI, thus forming complexes with proatherogenic and immunogenic properties. oxLDL/β2GPI complexes can localize inside the intima of the arterial wall and become a target of autoantibodies, thus promoting pro-inflammatory and pro-oxidant pathways and immune cell activation that contribute to the promotion of endothelial dysfunction and thrombotic events [24,25,26].

Patients with primary APS have increased levels of circulating oxidative stress-related markers when compared to healthy subjects. Of interest, APS patients with triple positivity for aPL, i.e., positive for LAC, aCL and aβ2GPI antibodies, show significantly higher levels of these markers than patients with single or double aPL positivity [27]. Different mechanisms have been identified as responsible for the induction of oxidative stress in APS. In particular, aPL antibodies have been indicated as a trigger for an increased oxidative status. A previous study by Perez-Sanchez et al. demonstrated that monocytes and neutrophils obtained from APS patients had increased levels of peroxide, NRF2 and antioxidant enzymatic activity, decreased levels of intracellular glutathione and altered mitochondrial membrane potential [28]. Of note, aCL antibodies levels positively correlated with the percentage of cells showing depolarized mitochondria and independently predicted the mitochondrial damage detected in monocytes from patients. An increase in peroxide levels and a decrease in both reduced GSH and nuclear NRF2 protein were observed in monocytes stimulated in vitro with IgG from APS patients, but not with IgG from healthy subjects. The treatment with antibodies from patients also increased the percentage of cells with depolarized mitochondria [28]. The preincubation of monocytes with antioxidants counteracted the peroxide levels’ increase and restored reduced GSH levels. Furthermore, the pretreatment of the monocytes with Coenzyme Q10, a component of the mitochondrial respiratory chain with antioxidant and anti-inflammatory properties, inhibited ROS production induced by the treatment with IgG from APS patients, thus indicating the mitochondrial electron transport chain as one of the main factors involved in the oxidative perturbation induced by APS autoantibodies [28].

A recent study demonstrated a significant association of anti-phosphatidylethanolamine autoantibodies (aPEs) with systemic ROS levels in APS patients with unexplained deep vein thrombosis [29]. These autoantibodies are directed against a neutral phospholipid, composed of esterified glycerol and phosphoethanolamine (PE), and have been demonstrated to improve the diagnosis of seronegative patients. As their main cellular targets are early endosomes, which play a pivotal role in intracellular trafficking, aPEs can affect various cellular processes. In particular, aPEs can dysregulate membrane redox enzymes and promote ROS-mediated signaling pathways, thus contributing to sustaining a pro-oxidant state in patients with APS.

Another mechanism implicated in oxidative stress is related to the interactions between aCL antibodies and circulating antioxidant enzymes, such as the paraoxonase-1 (PON1), an antioxidant enzyme that inhibits LDL oxidation. Patients positive for aCL autoantibodies have been demonstrated to show reduced activity of PON1 [30]. A cross-sectional study of 77 women positive for antiphospholipid antibodies and 77 controls confirmed these data demonstrating that PON1 activity was lower in women with aPL compared to the controls, and that they had greater structural vascular alterations, and impaired anti-inflammatory and antioxidant properties [31]. Moreover, aCL also promote oxidative stress by the alteration of nitric oxide (NO) and superoxide expression, thus sustaining the production of plasma peroxynitrite, a compound with pro-oxidant properties [32]. A clinical study demonstrated that levels of IgG aCL negatively predicted plasma nitrite (NO(2)(-)) and nitrate (NO(3)(-)) [33]. Furthermore, in another study, Ramesh and colleagues observed that aPLs obtained from human patients are able to reduce the plasma levels of NO metabolites by inhibiting eNOS activation, thus determining a decline in NO production, and promoting leukocyte adhesion to endothelial cells and thrombus formation [34].

Data obtained in vitro demonstrated that aPLs also promote oxidative stress by modulating the activity of the pro-oxidant enzymes of immune cells. In particular, it has been demonstrated that human monoclonal aPLs and IgG fractions from APS patients activate endosomal nicotinamide adenine dinucleotide phosphate (NADPH) oxidase and the generation of superoxide in plasmacytoid dendritic cells and monocytes. This activation determines the increased expression of toll-like receptor (TLR) 7 and TLR8 mRNA in plasmacytoid dendritic cells and in monocytes, respectively, and promotes inflammation, which is closely linked to pro-oxidant pathways [35].

A further mechanism by which aPLs promote oxidative stress is the direct activation of pro-inflammatory pathways. Autoantibodies specific for β2GPI are able to activate endothelial cells [36], platelets [37] and monocytes [38] and induce the activation of a complement cascade with a consequent release of procoagulation factors and inflammatory mediators, such as Tissue Factor, cytokines and metalloproteases [39]. This pro-inflammatory microenvironment promotes the production of ROS and reduces natural antioxidant defenses, thus promoting susceptibility to tissue damage. In this way, oxidative stress contributes to the immune response alteration and to the hypercoagulation state.

All these observations sustain the main role of autoantibodies in the induction of a pro-oxidant status in APS patients which, in turn, acts to promote atherothrombosis (Figure 1).

## 3. Altered Oxidant–Antioxidant Balance in APS Patients: New Biomarkers for Thrombosis Risk Evaluation

Low-to-moderate levels of ROS have important biological functions, as they contribute to the regulation of signal transduction and normal physiological processes. Endogenous antioxidant defense mechanisms play a role in maintaining redox homeostasis [40]. Oxidative stress occurs when there is an imbalance between ROS production and cellular antioxidant defenses, and the overproduction of ROS leads to biomolecules damage, particularly in lipids, proteins and DNA [40]. Experimental and clinical studies have demonstrated an alteration of the oxidant–antioxidant balance in APS and an increase in oxidative stress-related molecules has been observed in patients with APS. Severe combined immunodeficiency (SCID) mice producing aCL and aβ2GPI had reduced levels of NO and PON1 activity, as well as increased levels of superoxide and peroxynitrite [32]. A more recent study demonstrated that APS mice expressed increased levels of 47phox in the liver; this protein plays a role in the promotion of NADPH oxidase activity, thus sustaining a pro-oxidant state [41] (Figure 1).

Similar results were obtained in clinical studies. In particular, the presence of high levels of prostaglandin F2-isoprostanes have been detected in plasma from APS patients. These compounds derive from ROS-mediated arachidonic acid oxidation. As F2-isoprostanes are related to lipid peroxidation, they represent an endogenous marker of oxidative stress and have the ability to predict cardiovascular events [42]. A cross-sectional study demonstrated that aPL-positive patients had higher levels of isoprostanes, accompanied by higher levels of monocyte Tissue Factor expression, when compared with aPL-negative subjects [43]. In another clinical study conducted on 45 APS patients, higher concentrations of 8-isoprostanes were detected in patients as compared to control subjects [27]. Furthermore, increased peroxide and nuclear NRF2 levels and reduced intracellular glutathione concentrations were observed in leukocytes obtained from APS patients, indicating the pro-oxidant status of these cells [28]. In the same patients, the total antioxidant capacity was significantly reduced, thus suggesting an impaired capacity to counteract ROS production.

As oxidative stress has a pivotal role in the pathogenic mechanisms of APS, particularly those related to atherothrombosis, the determination of circulating oxidative stress biomarkers could be useful for the risk assessment of vascular complications in patients with APS. A clinical study conducted in 140 patients with primary and secondary APS aimed to evaluate the relationship between oxidative stress and endothelial damage as a risk factor of thrombosis revealed altered levels of different oxidative stress markers in patients compared to controls. Lipid hydroperoxydes levels were predictive for the impaired flow-mediated dilation of the brachial artery, thus suggesting their possible role as markers of endothelial damage [44]. However, in a more recent study aimed to assess the possible association between oxidative stress and thrombosis in primary APS, Vaz and colleagues did not observe any association between oxidative stress markers, namely plasma total antioxidant capacity, malondialdehyde (TBARs), carbonyl protein and 8-isoprostane, and the occurrence of arterial thrombosis [39]. Of note, they did not detect significantly different levels of these markers between patients and controls, suggesting that these contrasting results could be due to the low-grade inflammation occurring in patients with primary APS, which determines a mild oxidative state, not detectable by circulating oxidative stress markers.

These conflicting results call for further studies to determine the association of circulating markers of oxidative stress with the occurrence of thrombosis and their possible usefulness as predictive markers of risk.

## 4. The Role of Oxidative Stress in the Post-Translational Modifications of Antigens Associated with APS

The production of ROS at physiological levels cooperates with the resolution of inflammation and the maintenance of homeostasis in tissues [45]. An overproduction of ROS and/or a deficiency of the antioxidant machinery can cause a biochemical imbalance and consequent tissue damage [46]. The pathogenic roles of oxidative stress and inflammation in APS are also related to their ability to effect protein structural modifications. Pro-oxidant conditions may cause conformational changes in protein structures by promoting post-translational modifications (PTMs) [24] (Figure 1). Among PTMs, some can be directly affected by tissue ROS, by inflammatory microenvironments and/or by environmental factors; others, such as acetylation, glycosylation, phosphorylation and citrullination, can be influenced by more indirect downstream pathways affected by ROS [47].

PTMs are reversible or irreversible chemical reactions, mainly catalyzed by enzymes that occur in specific amino acids of a certain amount of proteins after their biosynthesis. These processes have a significant impact on the structure and function of proteins; indeed, these modifications play a key role in regulating the folding of proteins, their targeting to specific subcellular compartments, their interaction with ligands or other proteins and, eventually, their immunogenic properties [48,49]. PTMs can influence cellular processes and consequently, their dysregulation is related to the etiopathogenesis of numerous diseases, in particular, a variety of autoimmune responses depend on PTMs of self-proteins representing a potential trigger of several autoimmune diseases [45]. PTMs have been proposed as a link between the environmental factors and oxidative stress inflammation. Environmental factors (smoking, air pollutants, obesity and infections) are associated with PTMs in proteins, and it has been reported as a clear risk factor for breaking tolerance to multiple autoantigens [46,50].

The known PTMs in APS mainly involve β2GPI. β2GPI is a plasma glycoprotein that consists of five domains rich in cysteines, the V domain is critical for the binding of β2GPI to anionic phospholipids [47]. β2GPI plays an important role in the regulation of the blood coagulation system. It is involved in several biological processes, as well as coagulation, fibrinolysis, angiogenesis, thrombosis, autoimmune disease and pregnancy complications. β2GPI may be subjected to PTMs, including oxidation, carbamylation, glycosylation, phosphorylation and acetylation that can have a significant impact on the structure, stability and function of β2GPI in the adoption of an open configuration of the protein that may expose the cryptic epitope and facilitate autoantibody binding. The PTMs directly influence the function of β2GPI and contribute to an increase of its immunogenicity [46,51]. Thanks to the recent discoveries on the effect of oxidative PTMs that occur under conditions of increased oxidative stress, the study of the role of β2GPI continues to be explored [47].

In particular, PTMs of cysteines include the addition of oxygen. The oxidative/reductive conversion of β2GPI may contribute to the regulation of its biological activity. One of the most functional groups in various proteins is the free sulfhydryl group (SH) contained in the amino acid cysteine. Indeed, the majority of circulating β2GPI is present in a form containing unpaired cysteines (free thiols), which constitutes the reduced form of β2GPI. Sulfhydryl modifications can alter the function of proteins containing cysteines within their catalytic centers or at the protein–protein interaction interface. In an oxidative stress condition, an overproduction of ROS quickly reacts with cysteine residues, especially redox active cysteines, to form reversible or irreversible oxidized forms. The total amount of β2GPI and the relative amount of oxidized β2GPI is increased in APS patients with a thrombotic history [47,51].

Oxidative stress, together with inflammation, is also involved in nonenzymatic glycosylation (glycation of lysine residues) which is a non-enzymatic process that leads to the formation of early, intermediate and advanced glycation end products; these products can modify the structure and function of self-molecules. Elevated levels of antibodies to glucose-modified β2GPI were reported in APS patients. Several factors can modulate this process [51]. These products pile up in inflamed tissues and, by interacting with the receptor for advanced glycation end products, RAGE, that are expressed on the surface of vascular and immune cells, take part in the progression of inflammatory and immune-mediated diseases [52]. The modifications of self-structures may cause the formation of neo or cryptic epitopes that can be recognized by immune cells, thus promoting the activation of autoimmune responses. For that reason, oxidative stress and inflammation contribute significantly to the pathophysiology of APS by promoting the appearance of neoantigens that sustain the activation of autoimmune reactions and the occurrence of complications in this disease [51].

Another important PTM, related to inflammation and oxidative stress, is carbamylation, a non-enzymatic reaction of cyanate with the primary amine of lysine residues from proteins that generate homocitrulline [23]. Particular conditions, such as uremia, inflammation and cigarette smoking, can increase cyanate levels, which are responsible for the formation of these modified proteins [53]. While carbamylation has been related to protein aging and other pathological conditions, recent studies have identified carbamylated IgG proteins in patients with rheumatoid arthritis and this suggests that the presence of these antibodies may be useful to predict higher disease activity and may be related to inflammatory biomarkers [23,54]. Carbamylation occurs in an inflammatory milieu also by neutrophil infiltration and/or in a neutrophil-extracellular traps (NET)/NET-like release [55]. Indeed, during inflammation, polymorphonuclear neutrophils and macrophages can release myeloperoxidase (MPO), an enzyme that is stored in azurophilic granules. MPO contributes to the conversion of thiocyanate (derived from food or smoking) as a substrate with H_2_O_2_ to cyanate, thereby further increasing carbamylation [53]. In APS patients, inflammation, NETosis and oxidative stress may represent the main mechanisms able to induce β2GPI carbamylation [46]. In a recent study, Carb-β2GPI was identified as a ‘new’ antigenic target in APS. Carb-β2GPI is able to activate dendritic cells, inducing the up-regulation of CD80, CD86 and CD40, the activation of ERK, p38 MAPK and NF-κB and the release of IL-12p70. The serological results showed that a significant proportion of APS patients were positive for anti-Carb-β2-GPI. Moreover, since patients who tested positive for anti-Carb-β2-GPI reported a high risk of thrombocytopenia, this test may be considered a suitable approach in the clinical evaluation of seronegative (SN)-APS [55].

Although β2GPI is the most well-known protein in APS, several other proteins were described as antigenic targets and found to play a pathogenetic role in the syndrome. In particular, vimentin contains arginine residue that can undergo PTM to citrulline, catalyzed by the enzyme peptidyl arginine deiminase (PAD) which is active in different cell processes such as in NETosis. Recently, Alessandri et al. have identified citrullinated vimentin as a new autoantigen for aPL. They hypothesized that citrullination influences the production of specific autoantibodies by inducing conformational changes in vimentin. Antibodies specific for mutated citrullinated vimentin have been found in 26.6% of patients with APS. These results indicate a role for citrullination in the pathogenesis of APS and in association with the development of joint involvement [56].

Other PTMs, including phosphorylation and acetylation, may affect β2GPI antigenicity [23], in particular, lysine residue acetylation via acetyltransferases is one of the most common PTMs in proteins, and lysine acetylation has also frequently showed an impact on immune system regulation. After lysine acetylation, the majority of proteins, including β2GPI, are in an open conformation [57]. Finally, phosphorylation is a reversible PTM, it is formed by adding a phosphate group from ATP to the side chains of amino acids by kinases. Phosphorylation modifications occur most commonly on serine, followed by threonine and tyrosine residues [58].

## 5. The Role of Oxidative Stress in the Activation of Multiple Signaling Pathways in Autoimmune Diseases

Oxidative stress is the consequence of the balance alteration between ROS production and a defective detoxification, leading to the alteration of various signaling pathways and multiple biological processes through modifying proteins. These changes in the cellular functions may promote a dysfunctional immune response, inflammation, the induction of apoptosis, deregulation of autophagy, impairment of mitochondrial functions and many other mechanisms. Oxidative imbalance plays a role in the breakdown of immunological tolerance contributing to pathogenesis, pathological progression and the exacerbation of the symptoms in several autoimmune diseases [59,60] (Figure 1).

One of the mechanisms by which oxidative stress can trigger an inflammatory response in the pathogenesis of autoimmune diseases concerns the activation of the enzyme poly (ADP-ribose) polymerase-1 (PARP-1) [61]. PARP-1 belongs to enzymes involved in DNA damage sensing and repair and it activates following extensive DNA damage caused by excessive ROS formation [62].

PARP-1, through NF-κB activation, may induce the production of inflammatory (TNFα, IL1β and others) and effector T cell cytokines (IL4, IL5), as well as of inflammatory mediators such as metalloproteinases (MMP9), inducible nitric-oxide synthase (iNOS), several chemokines, prostaglandins (PGE2) and alarmins (HMGB1) [63,64,65]. On the other hand, PARP-1 modulates the inhibitory cytokine IL10 and the expression of Foxp3, a transcription factor required for regulatory T cell differentiation and function [66,67].

Metabolic cues such as oxidative stress trigger, in most cells, the redox-dependent activation of a mammalian target of rapamycin (mTOR), using a process that involves the cysteine oxidation of Rheb78 and raptor (regulatory-associated protein of mTOR) [68,69]. In turn, mTOR uses two interactive complexes—mTOR complex 1 (mTORC1) and mTOR complex 2 (mTORC2)—to execute cell-type-specific commands for growth, proliferation and survival [69,70]. In addition, the activation of mTORC1 can promote the proinflammatory skewing of T cell development and inhibit the expression of FoxP3 and the functions of T regulatory (Treg) cells [71,72].

Regarding this, oxidative stress plays a role as a regulatory checkpoint in the pathogenesis of systemic lupus erythematosus (SLE) and other autoimmune diseases, where the activation of mTORC1 has been described as a central pathway. In fact, the use of N-acetylcysteine (NAC), an amino acid precursor of glutathione treatment, also reversed the prominent activation of mTORC1 in double-negative T cells [73,74,75].

Oxidatively modified proteins, which act as neoantigens, can trigger pathogen-associated molecular patterns (PAMPs) and damage-associated molecular patterns (DAMPs) of the immune system that represent an important link between some environmental factors such as oxidative stress and autoimmune diseases [76]. In this context, as a response to oxidative stress, TLR-4 is activated, a widely studied TLR family member with a key role in the recognition of PAMPs and/or DAMPs and thus in the triggering of inflammation/autoimmunity responses. TLR-4 initiates a signaling pathway through Myeloid differentiation primary response 88 (MyD88) adaptor protein which, by interacting with the interleukin-1 receptor (IL-1R)-associated kinase (IRAK), leads to the activation of transcription factors including NF-κB [77].

A condition of oxidative stress can also activate heat shock factors (HSF) and induce a subsequent biosynthesis of heat shock proteins (HSPs), stress-induced cellular molecules that cooperate with the antioxidant system to inhibit or neutralize the cellular effects of ROS [78]. Moreover, HSPs are involved in regulating ERK1/2 in the MEK-ERK (mitogen-activated protein kinase) pathway, as well as interacting with Akt, in order to protect cells from apoptosis by the formation of the Akt-HSP complex [79,80]. Therefore, HSP proteins are found in association with the inflammation process and are able to activate immune regulatory mechanisms, including the expansion of Treg cells and/or the T helper 2 (Th2) cell population and, consequently, an arrest of the polarization of the pro-inflammatory T helper 1 (Th1) cell population [81,82]. For these reasons, the use of HSP is arousing interest in the context of autoimmune disease therapy, with the aim of inducing immunoregulatory mechanisms and preventing the uncontrolled activation of effector cell populations [83]. New pathophysiological mechanisms about the relationship between HSPs and oxidative stress include the activation and the comodulation of two transcriptional cytoprotective pathways, such as the Kelch-like ECH-associated protein 1 (Keap1)/NRF2 and HSP90/HSF [84].

## 6. The Interplay among Oxidative Stress, Inflammation and NRF2 Pathways in APS

APS is characterized by uncontrolled inflammation and the over-production of ROS. Increasing evidence supports a link between OS and inflammation in autoimmune diseases [48]. Kadl and colleagues showed, in TLR2-deficient mice treated with oxidized phospholipids, how an accumulation of oxidative tissue damage creates a microenvironment that causes “sterile” TLR2-dependent chronic inflammation [85]. Recently, Wang G et al., using lupus-prone MRL+/+ mice, demonstrated that trichloroethene (TCE) exposure accelerated an autoimmune response by inducing the dysregulation of TLR signaling and the concomitant impairment of NRF2 and its target gene HO-1. Antioxidant supplementation clearly ameliorated TCE-induced reduction in both NRF2 and HO-1 [86]. Antioxidants play a central role in the oxidative stress response and NRF2 activation is one of the main signals studied to counteract excessive ROS production and influence disease outcome, including autoimmune diseases. NRF2 is considered crucial in the redox switch due to its regulation of about 1% of human genes via an antioxidant response element (ARE). It is a key redox-sensitive transcription factor which deals with the transcription of more than 500 genes that are involved in redox balancing, xenograft reactions, metabolism and cell survival [87]. The target genes comprise phase I and II detoxification enzymes, transporters, growth factors and other transcription factors. The downstream genes NQO1, HO-1 and GCLC exert antioxidant effects and instead, the downstream targets such as TGF-β and NF-κB are related to inflammation. In fact, multiple pathways triggered by oxidative stress downstream activate NK-κB, which may also lead to the suppression of NRF2 by competing for the transcriptional co-activator CREB-binding protein-p300 complex [86,88]. Therefore, NF-κB behaves as a link in redox-sensitive inflammatory pathways. It can be considered a key molecule in cellular protection against oxidant-mediated damage, with a regulatory role between oxidative stress and the battery of antioxidant and cytoprotective proteins controlled by NRF2 [19,89,90].

In addition, the NRF2/ARE signaling pathway may be considered the primary pathway for intracellular redox balance and the molecules that it induces exert antioxidant and anti-inflammatory effects by upregulating various cytoprotective enzymes and proteins. NRF2 is physiologically localized in the cytoplasm and anchored to the intracytoplasmic actin cytoskeleton in the form of the Keap1–NRF2 complex. E3 ubiquitin ligase promotes the ubiquitin-mediated degradation of NRF2 and the maintenance of a basal steady-state level of NRF2 activity. Otherwise, under oxidative or electrophilic stress conditions, NRF2 dissociates from Keap1 and translocates to the nucleus forming a heterodimer with the small musculoaponeurotic fibrosarcoma (sMAF) proteins and binds to an Antioxidant Response Element (ARE) in the promoter region of cytoprotective genes [87]. In addition to regulation by Keap1, the NRF2 pathway is activated by kinases that play a key role in the phosphorylation-mediated activation of NRF2. In particular, the phosphorylation of NRF2 at Ser 40 by PKC, resulting in the dissociation of NRF2 from the Keap1–NRF2 complex stabilization and newly transcribed NRF2, translocates into the nucleus. Moreover, it has been reported that NRF2 can be activated by PI3K, JNK, ERK and Akt signaling [88,91,92].

Interest in the cytoprotective role of NRF2 is growing as a potential drug target in APS, in which OS and inflammation underlie the disease’s pathogenesis. In patients with APS, the NRF2 activation was repressed in monocytes, thus suggesting a defect in the NRF2 signaling pathway [28]. In an animal study, mice lacking NRF2 developed several characteristics of the autoimmune-mediated lesions, similar to those of human SLE, the spontaneous development of autoantibodies, increased T-cell proliferation, multi-tissue inflammatory lesions, along with glomerulonephritis, the intravascular deposition of immunoglobulin complexes and premature death [93]. Analyses of antioxidant-induced gene expression showed that the NRF2 knockout mice had a reduced expression of the NRF2 target genes HO-1 and NQO1 and concomitantly displayed increased oxidative lesions in multiple tissues, thus confirming an interplay between the NRF2 function, TLR signaling and autoimmune surveillance in the development of autoimmune disorders [86,94,95].

Keeping in mind that an unbalanced accumulation of ROS has been linked to the occurrence and development of many acute and chronic diseases, including APS, pharmacological regulation of NRF2 pathways is an emerging treatment strategy. However, APS is a complex pathological process, in which multiple signaling pathways are involved in the immunopathogenesis of the disease. Thus, a thorough knowledge of the interplay among oxidative stress-responsive redox signaling pathways, including TLRs signaling, NRF2 and NF-κB, may contribute to a better comprehension of the immunopathogenesis of APS, driving towards a personalized medicine.

## 7. Effects of Food and Dietary Supplementation on Oxidative Stress in APS

Nowadays, there is an increasing interest in nutritional interventions because of their potential role in preventing tissue damage related to uncontrolled inflammation and oxidative stress. The presence of aPLs is often associated with tissue damage, such as vascular events, miscarriages, cerebral or dermatologic manifestations, probably because of the direct interaction of aPLs with tissues [96]. More so, the occurrence of ischemia/reperfusion (I/R) injury may lead to enhanced aseptic inflammation and the stimulation of oxidative stress. In healthy physiological conditions, this excessive ROS accumulation is kept down by hydro- and liposoluble anti-oxidants such as glutathione and liposoluble vitamins, as well as antioxidant enzymes. In the pathophysiology of APS a failure in the physiological response to ROS and inflammation has been extensively documented [23,55,97]. The chronic nature of autoimmune diseases such as APS and the associated high burden of monitoring and treatment emphasizes that there is a need for novel approaches to care for APS patients. In recent years, supplementation with natural compounds has clarified its role as a possible supporting therapy for their numerous positive pleiotropic effects, including the amelioration of endothelial dysfunction and arterial stiffness, as well as anti-inflammatory and anti-oxidative properties [98]. In addition, a relatively large amount of epidemiological and clinical data encourage the tolerability and safety of many nutraceuticals. Recently, Nocella and co-authors exhaustively provided an overview of potential therapeutic approaches based on antioxidant supplementation to blunt oxidative stress and to prevent atherothrombotic complications in APS patients [24]. Omega-3 polyunsaturated fatty acid (n-3 PUFA), vitamins, CoQ10 and polyphenols are the major natural compounds investigated in the murine experimental model of APS, in humans and in in vitro experimental studies. However, the lack of data from sufficient randomized controlled trials slows down the process of antioxidant supplementation becoming a potential management option in APS.

Many natural compounds have been reported to control oxidative stress by regulating redox homeostasis and inflammation and leading to cell survival, neovascularization and the repair of damaged tissues by promoting the activation of the transcription factor NRF2 [99]. More than 1000 genes possessing an Antioxidant Response Element in their promoters can be activated by NRF2 during oxidative stress [100,101]. As reported above, many experimental studies have shown the protective role of the NRF2 pathway both in physiological conditions and in a wide array of disease models [102], thus making this pathway a valuable therapeutic target in chronic diseases marked by chronic low-grade redox alterations and inflammation. In patients with APS, NRF2 activation was repressed in monocytes, thus suggesting a defect in the NRF2 signaling pathway [28]. In the same study, the authors demonstrated the beneficial effects of CoQ10 in vitro in the prevention of mitochondrial dysfunction and oxidative stress and in the suppression of the expression of prothrombotic markers relevant to the pathophysiology of APS activity. A recent study was conducted by Lin et al. [103] to determine the therapeutic effects of Artemisinin anologue (SM934), a plant compound employed in Chinese traditional medicine, in a murine model that resembled human APS associated with SLE. This treatment prevented autoantibodies’ production and decreased inflammatory cytokine production and oxidative stress in the kidney, suggesting the enhancement of NRF2 signaling and the expression of its target genes. Recently, Ryenga et al. reported a beneficial effect still related to NRF2 activation in APS [104]. By using the anti-inflammatory and antioxidant flavonoid taxifolin as a dietary supplement in a murine model of APS thrombosis, the authors described an effective way to target NETs, important players in APS. Indeed, taxifolin attenuated disease-relevant activities such as autoantibody formation and large-vein thrombosis by activating the NRF2 pathway.

To date, several bioactive compounds have been identified as “NRF2 activators” and they mediate protection against oxidative stress and inflammation-derived injury, thus representing new therapeutic strategies for tissue healing and the prevention of the recurrence of tissue damage. Since, in APS patients, the physiological response to ROS and inflammation is not always sufficient to provide full protection, and results show that the administration of redox-targeted supplements (e.g., by nutraceuticals) could improve remission in terms of circulating prothrombotic markers, an adjunctive therapy addressing the NRF2 pathway could be a promising therapy (Figure 2). However, further studies are needed to investigate the role of NRF2 in APS patients, especially to determine the level of NRF2 activation that would significantly slow disease progression when various NRF2 activators are used. In silico studies, cellular and animal models should be established to test the efficacy and side effects of NRF2 activators.

Among various pathophysiological processes that play a role, oxidative stress is considered a key mechanism driving the onset and progression of APS. Oxidative stress is defined as an imbalance between oxidants and antioxidants in favor of the oxidants, leading to a disruption of redox signaling and control and/or molecular damage. Patients with APS are characterized by disturbances of redox signaling pathways, which can be quantified by measuring components of the systemic redox status, i.e., redox biomarkers. The discovery and validation of redox biomarkers, particularly those related to thrombosis, may represent a useful tool in the clinical management of patients with APS, but as of yet, it is not always clear what all these biomarkers exactly represent and what their relative contributions are in relation to the whole-body redox status. Considering that supplementation with natural compounds have shown their role as possible supporting therapies for their numerous positive pleiotropic effects, personalized redox medicine approaches are emerging in the context of APS. A broad range of NRF2 activators or the dietary intake of antioxidant-rich food with the aims of activating endogenous antioxidant defense systems, represent an adjunctive therapy for restoring redox homeostasis and for improving remission in terms of circulating prothrombotic markers.

## 8. Conclusions

This review deals with the role of oxidative stress in the initiation and progression of APS. Large evidence points out the role of oxidation as a key regulatory checkpoint of “antiphospholipid antibody” production, by promoting the post-translational modifications of proteins and/or triggering multiple signaling transduction pathways. Among these, the protective role of the NRF2 pathway has been elucidated. This knowledge provides new insights into the pathogenesis of APS and introduces new hypotheses for valuable therapeutic targets, including personalized redox nutraceutical medicine approaches.

## Figures and Tables

**Figure 1 biomolecules-13-01221-f001:**
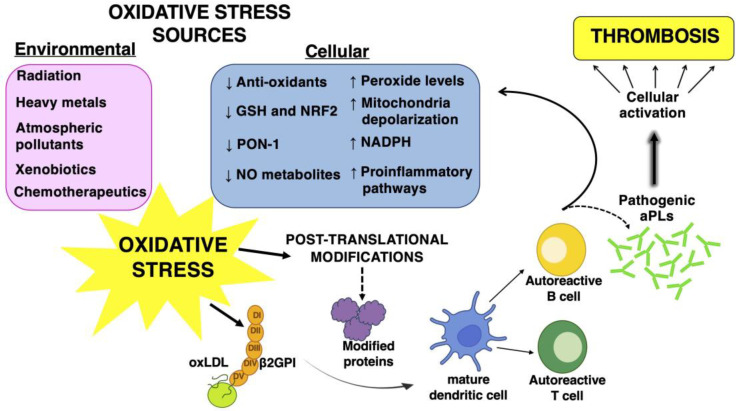
Schematic representation of the connection between the induction of a pro-oxidant status in APS patients and the antibody production (aPLs).

**Figure 2 biomolecules-13-01221-f002:**
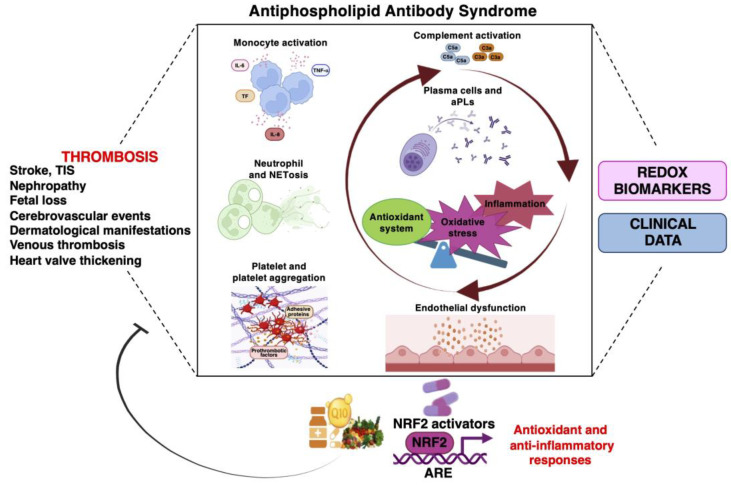
Dietary intake of antioxidant-rich food and/or NRF2 activators as interventions to target oxidative stress imbalance and inflammation in APS.

## Data Availability

Not applicable.

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
