# Peer review of "Oxidative Stress as a Regulatory Checkpoint in the Production of Antiphospholipid Autoantibodies: The Protective Role of NRF2 Pathway"

_biomolecules, 2023, doi:10.3390/biom13081221_

Round 1

Reviewer 1 Report

This is a comprehensive review on the importance of oxidative stress in APS. The paper is well balanced and addresses the most important issues regarding this subject.

Line 131 - there is a typo: "autoanti-bodies"

Author Response

We thank the reviewer for her/his useful suggestion.

Reviewer 2 Report

This manuscript reviews the role of oxidative stress in APS and its impact on the production of antiphospholipid autoantibodies. It discusses the involvement of oxidative stress in APS progression, the biomarkers of thrombosis risk, and the post-translational modifications of antigens associated with APS. The review also highlights the potential of food supplements and NRF2 activators as therapeutic approaches to reduce APS-related complications.

This manuscript is overlapped with the previous authors’ review paper.(https://www.ncbi.nlm.nih.gov/pmc/articles/PMC8615138/)

In this manuscript, the focus on NRF2 and antibody production is different from the previous paper, but the first half of the paper is similar to what they wrote before without mentioning NRF2. Also, although Figure 1 shows the pathogenesis, clinical manifestations, and future prospects of APS with a focus on NRF2, platelets and complement are not mentioned in the text.

Many parts of the text have nothing to do with the title or aim. Moreover, comprehensive details concerning the interplay between NRf2 and the pathogenesis of APS are absent.

Specific comments

1.     Given the focus on antibody production, it would be pertinent to include a figure elucidating the connection between antibody production and oxidative stress.

Author Response

 In this manuscript, the focus on NRF2 and antibody production is different from the previous paper, but the first half of the paper is similar to what they wrote before without mentioning NRF2.

Reply: We agree with Reviewer’s observation about the topic overlapping, however we attempted to highlight how oxidative stress is a driver in the APS pathogenesis as a consequence of a balance alteration between ROS production and defective de-toxification, leading to abnormal signaling pathways, aPLs formation and tissue injury. All these topics are matter of our investigation and need to be reviewed to drawn attention for this new therapeutic field in the treatment of APS.

Also, although Figure 1 shows the pathogenesis, clinical manifestations, and future prospects of APS with a focus on NRF2, platelets and complement are not mentioned in the text.

Reply: As suggested by the reviewer, we have mentioned platelets and complement in the text in accordance with the figure 1, now figure 2 (lines 175-176).

Many parts of the text have nothing to do with the title or aim.

Reply:  We thank the reviewer for her/his useful suggestion. We have rephrase the title with a more comprehensive one.

Moreover, comprehensive details concerning the interplay between NRF2 and the pathogenesis of APS are absent.

Reply: Following reviewer criticism, we have added a new paragraph about the interplay among oxidative stress, inflammation and NRF2 pathways in APS (lines 401-469).

Specific comments

  1. Given the focus on antibody production, it would be pertinent to include a figure elucidating the connection between antibody production and oxidative stress.

Reply:  We agree with Reviewer’s observation, we have added a new figure to elucidate this part (see Figure 1).

Reviewer 3 Report

In this manuscript, the authors propose an interesting review on the role of oxidative stress in the antiphospholipid syndrome (APS), in particular in the production of aPL, in the post translational modifications and in the regulation of signaling transduction pathways. They conclude on a potential redox nutraceutical medicine approach.

Some remarks and proposals can be made:

-           It would be interesting to add a first introductory part on the generation of ROS

-          The authors could add the link between “non conventional “ aPL and ROS, in particular concerning anti-phosphatidylethanolamine autoantibodies (aPE). Endosomal compartment is a source of ROS production, and it has been recently identified as the cellular target of these autoantibodies. More recently, a significant association between aPE positivity and systemic ROS production in patients which led us to hypothesize a new mechanism of action of aPEs in thrombosis through a signaling related to oxidative stress. J Clin Med. 2022 Feb 27;11(5):1297

Author Response

Some remarks and proposals can be made:

-           It would be interesting to add a first introductory part on the generation of ROS

Reply: Following reviewer criticism, we have added a new paragraph in the Introduction section on the generation of oxidative stress (lines 55-68).

-          The authors could add the link between “non conventional “ aPL and ROS, in particular concerning anti-phosphatidylethanolamine autoantibodies (aPE). Endosomal compartment is a source of ROS production, and it has been recently identified as the cellular target of these autoantibodies. More recently, a significant association between aPE positivity and systemic ROS production in patients which led us to hypothesize a new mechanism of action of aPEs in thrombosis through a signaling related to oxidative stress. J Clin Med. 2022 Feb 27;11(5):1297

Reply: Following the reviewer’s suggestion, we have now added the link between aPEs and ROS in the paragraph 2 “aPL as a trigger for a pro-oxidative state in APS patients” and we have added the relevant reference indicated by the reviewer (lines 140-148).

Round 2

Reviewer 3 Report

good revision